# Idiopathic Scoliosis Progression: Presenting Rib and Segmental Rib Index as Predictors—A Literature Review

**DOI:** 10.3390/medsci13020062

**Published:** 2025-05-21

**Authors:** Theodoros B. Grivas, Elias Vasiliadis, Konstantinos Soultanis, Marios Lykissas, Galateia Katzouraki, Nikolaos Sekouris, Dimitrios Lykouris, Christina Mazioti, Aristea Mamzeri, Despina Papagianni, Eleni Potamiti, Alexandros Kastrinis, Evangelos Theodosopoulos

**Affiliations:** 1Department of Orthopedics & Traumatology, “Tzaneio” General Hospital of Piraeus, 185 36 Piraeus, Greece; 23rd Department of Orthopaedics, School of Medicine, National and Kapodistrian University of Athens, KAT Hospital, 145 61 Athens, Greece; 31st Department of Orthopaedics, School of Medicine, National and Kapodistrian University of Athens, 124 62 Athens, Greece; ksoultanis@otenet.gr; 4Metropolitan Hospital, Ethnarchou Makariou 9 & El. Venizelou 1, Neo Faliro, 185 47 Pitaeus, Greece; lykissasm@gmail.com (M.L.); dimitrisluc@gmail.com (D.L.); 5Spinal Department of Hygeia Hospital, 4 Erythrou Stavrou, 151 23 Maroussi, Greece; gkatzouraki@hotmail.com; 61st Department of Orthopedics, P. & A. Kyriakou Children’s Hospital, 23 Levadeias, 115 27 Athens, Greece; nick_sekouris@yahoo.com; 7Office of Health Visitors, “Tzaneio” General Hospital of Piraeus, 185 36 Piraeus, Greece; maziotix@gmail.com; 8TOMY Attica Square, 104 45 Athens, Greece; mamzeri_aristea@hotmail.com; 9School Nurse—Health Visitor, Special Primary School of Rafina, 190 09 Rafina, Greece; papdes2009@hotmail.com; 10Department of Early Intervention & Rehabilitation General, Children’s Hospital of Athens “P. & A. Kyriakou”, 115 27 Athina, Greece; epotamiti@gmail.com; 11Health Assessment and Quality of Life Research Laboratory, Department of Physiotherapy, School of Health Sciences, University of Thessaly, 351 32 Lamia, Greece; alexkastrinis@uth.gr; 12Athens Scoliosis, Moschato, 183 45 Attica, Greece; etheodosop@yahoo.com

**Keywords:** idiopathic scoliosis, rib index, segmental rib index, progression factors, prediction factors, models of progression prediction

## Abstract

This report provides a concise selective representative overview of the predictor factors for progression in Idiopathic Scoliosis (IS). The Cobb angle method, rib hump deformity, imaging and advanced techniques for assessing skeletal maturity serve as key elements in evaluating prognostic factors for IS progression based on the patient’s age at diagnosis—particularly in Infantile Idiopathic Scoliosis (IIS), Juvenile Idiopathic Scoliosis (JIS), and Adolescent Idiopathic Scoliosis (AIS). The commonly used approaches for determining skeletal maturity include the assessment of the iliac apophysis and scoliosis curve deterioration, the Sanders skeletal maturity staging system, the distal radius and ulna (DRU) classification for predicting growth spurts and curve progression in IS, as well as the ossification of vertebral epiphyseal rings, the humeral head, and the calcaneal apophysis. Prognostic factors influencing IS progression are further discussed in relation to the patient’s age at onset—whether in infancy, childhood, or adolescence—as well as in both untreated and braced AIS patients. Additionally, the apical convex rib–vertebra angle in AIS is explored as an indicator of progression. Predictors for curve progression at skeletal maturity are outlined, along with various models for forecasting IS deterioration. Lastly, the Rib and Segmental Rib Index, a rib cage deformity parameter, is introduced as a predictor of scoliosis progression. In conclusion, this concise and selective overview of predictor factors for progression in IS highlights the current understanding of IS progression factors. It also introduces the Rib and Segmental Rib Index—a rib cage deformity parameter—as a predictor of IS progression.

## 1. Introduction

The information required to analyze the factors influencing the progression of idiopathic scoliosis (IS) is sourced from scoliosis clinics in hospitals and School Scoliosis Screening (SSS) programs. Within SSS programs, examiners frequently detect cases of mild or early-stage IS, enabling them to collect data from this specific group of children. Conversely, physicians in outpatient clinics primarily gather information from patients with more advanced scoliosis, at a stage suggesting that it may not have been captured during the initial or mild phases of the condition. This distinction in data collection is significant, as the underlying biomechanical characteristics of early-stage and mild scoliosis are likely different from those observed in severe cases. Furthermore, the identification of curves that will progress beyond the surgical threshold was historically a significant challenge of orthopedic surgeons who examine and treat children with mild and moderate scoliotic curves [1].

At the onset of IS and in its milder forms, genetic, epigenetic, and biological factors play a predominant role in its development, with minimal to no structural changes in the skeleton. However, the influence of patho-biomechanics, though initially secondary, becomes more pronounced as the condition progresses and skeletal deformities become well established [2].

Noshchenko et al. (2015) highlighted that inconsistencies across published studies, coupled with variations in assessment methods and clinical parameters, limit the reliability of existing criteria for predicting which children with mild IS will experience curve progression necessitating intervention. Furthermore, their review did not identify any predictive models for the progression of adolescent idiopathic scoliosis (AIS) that could be recommended as clinical diagnostic tools [3].

Commonly cited predictors in the literature include biological factors (such as sex, age, and growth patterns), anatomical and clinical characteristics (including thoracic deformity, rib hump, leg length discrepancy, and trunk imbalance), radiographic or imaging markers (such as the Cobb angle, vertebral rotation, torsion, tilt, wedging of the vertebral body, segmental thoracic ratios, rib–vertebra angles (RVAs), segmental RVAs, and the torsion index), as well as genetic influences (including specific genes and familial inheritance patterns). A multivariate logistic regression model which includes biomarkers, as well as clinical and radiographic findings, was recently introduced by Zhang et al. to predict curves that will possibly exceed a Cobb angle of 40 degrees [4].

In contemporary research, three-dimensional analysis is increasingly employed to examine IS curve morphology, as studies limited to coronal, sagittal, or transverse planes alone have inherent limitations. However, key radiological parameters—such as the Cobb angle, Mehta RVAs, and Perdriolle angles—are traditionally measured using posteroanterior (PA) and lateral radiographs. While 3D assessments of the spine and rib cage provide valuable insights, they require specialized imaging equipment, and 3D reconstructions from CT scans are not routinely performed due to radiation exposure concerns [5,6,7,8,9,10,11]. There are numerous efforts in the literature to create predictive models for curve progression but most of them proved to be ineffective mainly due to the fact that the weighting of each individual factor in these models is not known [12].

Recent years have seen a rise in interdisciplinary research applying machine learning to clinical data to develop predictive models for scoliosis progression. These studies often employ specialized terminology, which can be complex and difficult to interpret, yet the models themselves frequently rely on 2D parameters extracted from standard radiographs [13,14,15,16,17,18].

From a practical standpoint, studies utilizing PA radiographs continue to offer significant value. Lateral radiographs are not routinely obtained for children with IS, and many hospitals face limitations in conducting retrospective studies, often relying primarily on frontal-plane radiographs. Additionally, standard chest radiographs of children and adolescents, commonly archived in medical records, can be effectively repurposed for research without necessitating additional radiation exposure. A recent effort has explored generating sagittal radiographs from coronal views using a Generative Adversarial Network (GAN)-based deep learning framework for AIS. However, as the study’s authors acknowledge, “although these synthetic images appear visually similar to real ones, their quality is still inadequate for precise clinical evaluation” [19].

This review aims to summarize the parameters used in a representative selection of studies as predictors of scoliosis curve progression. Furthermore, it introduces the Rib Index and the Segmental Rib Index as potential predictors for IS progression.

## 2. Cobb Angle—Rib Hump

A normal spine consists of bony vertebrae which connect the base of the skull to the pelvis and are joined by cartilaginous intervertebral discs. The vertebral column is straight in the coronal plane and presents four curves in the sagittal plane, two kyphotic in the thoracic and in the sacral level and two lordotic in the cervical and in the lumbar segment. Any deviation of the spine in the coronal plane beyond 10 degrees is considered a scoliotic curvature and it usually also affects the sagittal curves, and eventually a 3D deformity of the spine occurs.

Since 1948, when John Robert Cobb introduced his technique for evaluating scoliotic curvature, the Cobb angle has been recognized as the benchmark for determining the severity of IS. Furthermore, it serves as the primary predictor in nearly all research studies and models related to IS progression (Figure 1, [20]).

The forward bending test, also known as the Adams test, was named after the English physician William Adams, who first described it in 1865 [21]. This test is commonly used as an initial clinical evaluation for children suspected of having IS. It is frequently employed in school scoliosis screening programs and can also be applied to patients with a family history of scoliotic posture or cases where scoliosis of uncertain origin is observed. Duval-Beauper (1992) reported that over 95% of patients with an initial supine angle exceeding 17 degrees, a standing angle greater than 24 degrees, and a rib hump larger than 11 mm experienced progressive scoliosis [22,23]. Additionally, in 95–100% of patients whose parameters at follow-up surpassed these thresholds (supine angle: 17 degrees, standing angle: 24 degrees, rib hump: 11 mm), the condition continued to worsen, confirming curve progression.

## 3. Methods for Assessing the Degree of Skeletal Maturity

Assessing skeletal maturity in AIS could significantly influence disease management. However, given that AIS is a complex and multifactorial condition, it is unlikely that a single metric will suffice to accurately predict its progression. As more contributing factors to IS progression are identified, it is crucial for the scientific community to collaborate in developing treatment strategies grounded in reliable and consistent algorithms [24].

Several methods have been utilized to determine skeletal maturity, including Risser’s method (Risser sign), the Sanders maturity scoring system, the distal radius and ulna (DRU) classification, ossification of the vertebral epiphyseal rings, ossification of the humeral head, and ossification of the calcaneal apophysis, as shown in Figure 2 [25,26,27,28,29,30].

### 3.1. The Iliac Apophysis and the Evolution of Curves in Scoliosis

Zaoussis and James (1958) carried out a study investigating the radiographic characteristics of the ossification center in the iliac apophysis and its correlation with other indicators of maturation, including the onset of menstruation and the formation of vertebral body apophyses in relation to scoliosis curve progression. This study was the first to present statistical evidence demonstrating that skeletal maturation, marked by the complete development of the iliac apophysis, signifies the conclusion of substantial scoliosis progression (Figure 3, [31]).

In 1984, Lonstein and Carlson presented research on the probability of curve progression in untreated AIS during growth, considering the Risser sign grade and curve severity (Figure 4, [32]).

### 3.2. Sanders Skeletal Maturity Staging System

The Sanders skeletal maturity staging system is a simplified approach to evaluating skeletal development. It is a reliable method that exhibits a stronger association with the progression of IS compared to the Risser sign or skeletal age assessment using the Greulich and Pyle method. Although it requires some initial learning, it is easy to implement in clinical settings. When combined with information about the curve type and size, this system effectively predicts the likely progression of IS curves over time [26]. Curve progression in IS showed a strong correlation with the initial curve magnitude and skeletal maturity in both female and male children. All patients at Sanders stage (SS) 2 with an initial Cobb angle of 25° or greater experienced progression, as did those at SS1 and SS3 with an initial Cobb angle of 35° or greater. Similarly, all patients with an initial Cobb angle of 40° or greater progressed, except those in SS7. On the other hand, none of the patients with an initial Cobb angle of 15° or less, or those in SS5, SS6, and SS7 with an initial Cobb angle of 30° or less, showed progression [26].

### 3.3. Distal Radius and Ulna Classification (DRU) Scheme in Predicting Growth Peak and Curve Progression in Idiopathic Scoliosis

The DRU (distal radius and ulna) classification by Luk et al. (2014) serves as a prognostic framework for evaluating growth potential in children and adolescents. It relies on radiographic assessment to determine skeletal maturity, categorizing development into 11 radius stages (R1–R11) and 9 ulna stages (U1–U9). Significant markers within this system include a Peak Growth Phase which is recognized at radius stage R7 and ulna stage U5, signifying a period of accelerated growth, and an end-of-growth phase which is represented by radius stage R10 and ulna stage U9, indicating that skeletal growth is approaching completion or has ceased. This classification has been validated as a dependable tool for forecasting growth stages, making it particularly valuable in clinical applications for growth-related interventions, such as those in IS [27].

### 3.4. Ossification of the Vertebral Epiphyseal Rings

The ossification center in the epiphyseal ring emerges between the ages of 11 and 14, appearing earlier in girls than in boys. This timeframe largely coincides with the period of curve progression in AIS [29].

### 3.5. Ossification of the Humeral Head

The proximal humeral ossification system provides a reliable method for estimating Peak Height Velocity (PHV) in individuals with IS using standard spine radiographs, thus removing the need for separate hand radiographs to assess bone age. This method improves the precision of maturity predictions, enabling healthcare providers to more accurately assess a patient’s skeletal maturity in relation to PHV. As a result, it supports more informed treatment decisions while minimizing additional radiation exposure, time, and costs. Assessing the proximal humeral physis in spine radiographs that include the shoulder serves as a practical and valuable tool for determining skeletal maturity in adolescents with IS; also, it was reported that the humeral head classification system showed a strong relationship with age at PHV and the remaining growth percentage. Furthermore, the staging system demonstrated excellent reliability in both inter-observer and intra-observer assessments, suggesting its broad applicability [33]. Additionally, Li et al. (2018), presented a formula to assess the progression to a surgical range of an AIS curve and the score is calculated by subtracting from the Cobb angle the tenfold stage of ossification of the humeral head, as shown in Figure 5 [33].

### 3.6. Ossification of Calcaneal Apophysis

The calcaneal apophysis undergoes ossification in a well-defined sequence, divided into six stages. These stages are closely tied to specific periods related to the PHV, making the calcaneal system a reliable method for assessing skeletal maturity. The PHV occurs before the iliac bone begins to ossify, while the calcaneal apophysis follows a sequence of four ossification stages before PHV and two stages after it [34].

Ossification of the calcaneal apophysis is an effective indicator of skeletal maturity during adolescence. The calcaneal system is most useful in pinpointing maturity in the five years leading up to PHV, whereas the Sanders system is more effective for identifying maturity after PHV. By combining different maturity systems, it is possible to achieve a more accurate assessment of maturity than relying on any single system alone [30].

## 4. Prognostic Factors of IS Progression According to the Patient’s Presentation Age

After evaluating a child or adolescent with scoliosis, both the parents and the patient often inquire about the prognosis of the condition. A useful way to categorize the prognosis is based on the patient’s age and presentation, which includes Infantile Idiopathic Scoliosis (IIS), Childhood Idiopathic Scoliosis (CIS), Adolescent Idiopathic Scoliosis (AIS) in its untreated form, AIS patients who have been treated with braces, and those with Skeletally Mature Idiopathic Scoliosis (IS).

### 4.1. Infantile Idiopathic Scoliosis (IIS)

IIS can be categorized based on the resolution or progression of the spinal curve into several types: early resolving, late resolving, benign progressive, malignant progressive, and dysplastic. It is more commonly seen in boys, particularly in those with left thoracic or thoracolumbar curves. A distinctive feature of IIS is that it is the only form of IS known to sometimes resolve on its own, improving or even disappearing without the need for treatment, except for monitoring.

In 1930, Hartenstein [35,36] reported that spontaneous correction could occur without intervention, although it was not possible to predict this at the time of diagnosis. Mehta, in 1972, introduced the rib–vertebra angle (RVA) as a tool to differentiate between resolving and progressive IIS. Her study found that when the RVA difference (RVAD) was of less than 20°, around 80% of patients experienced spontaneous resolution of their scoliosis. However, in patients with an RVAD greater than 20°, 80% showed progression of their scoliosis [37].

A study by Agadir et al. (1992) followed 18 patients with progressive IIS over an average of 7 years. It showed that the relationship between the progression of the spinal curvature (measured by Cobb Angle, CA) and apical vertebral rotation (AVR) could identify four types of progressive IIS curves. Type I showed synchronous progression of CA and AVR, Type II showed progression of CA, while AVR resolved to some extent, Type III showed resolution of CA, while AVR continued to progress, and Type IV showed temporary resolution of both CA and AVR before they progressed again [38]. This dissociation between CA and AVR progression suggests complex neuromuscular factors that regulate scoliosis development in both the frontal and transverse planes [38,39].

Progressive IIS, like AIS, may result from asymmetries in the central pattern generators (CPGs) controlling trunk movements during gait [40,41]. According to Mehta (1972), progressive IIS occurs when the apical RVAD exceeds 20° during and is at phase 2 [42]. Kristmundsdottir et al. (1985) found that a convex RVA measurement was as reliable as the RVAD in predicting prognosis, with an RVA under 68° on the initial radiograph typically indicating curve progression [43].

In studies by Grivas et al. (1990, 1991, 1992, 2006), radiological assessments of the spine and rib cage showed that children with progressive IIS had narrower rib cages compared to controls, with a funnel-shaped upper chest, as shown in Figure 6 and Figure 7. They also found that vertebral rotation at the upper limit of the thoracic curve could predict the progression of IIS, suggesting that neuromuscular factors contribute to the condition [44,45,46]. Additionally, in surgically treated IIS cases, preoperative counter-rotation of the T4 vertebra predicted apical vertebral rotation at follow-up, and a decrease in vertebral tilt from T5 to T1 in the upper chest was observed on preoperative PA radiographs. The maximum RVAD of 50° was found at T6, above the apical vertebra [41,42,43,44,45,46].

Perdriolle and Vidal (1985) [47] stated that Specific Rotation (SR) serves as an indicator of IIS progression. This refers to the total of the two rotational angles observed in the vertebrae adjacent to the upper-end vertebra. IIS is likely to progress if the SR exceeds 5 degrees at the age of 2, 10 degrees at the age of 4, and 20 degrees at the age of 6 [47], as shown in Figure 8.

**Figure 6 medsci-13-00062-f006:**
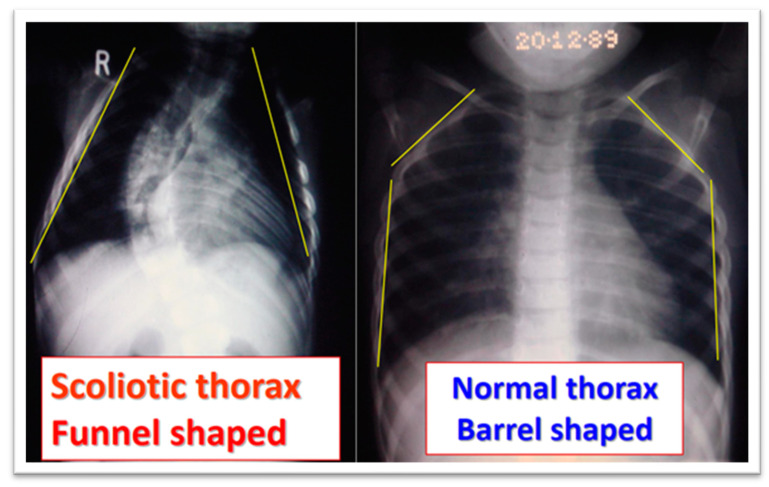
A radiograph of a boy aged 1 year 7 months with progressive IIS and that of a normal peer [44,45,46,48,49,50]).

**Figure 7 medsci-13-00062-f007:**
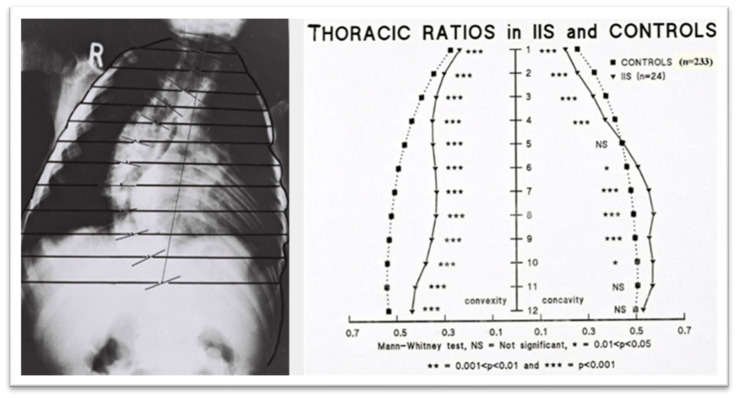
The thoracic ratios for the sample of patients with IIS compared with controls. The comparison shows that the scoliotic thorax is significantly narrower than that of the controls at all spinal levels (modified from [45]).

**Figure 8 medsci-13-00062-f008:**
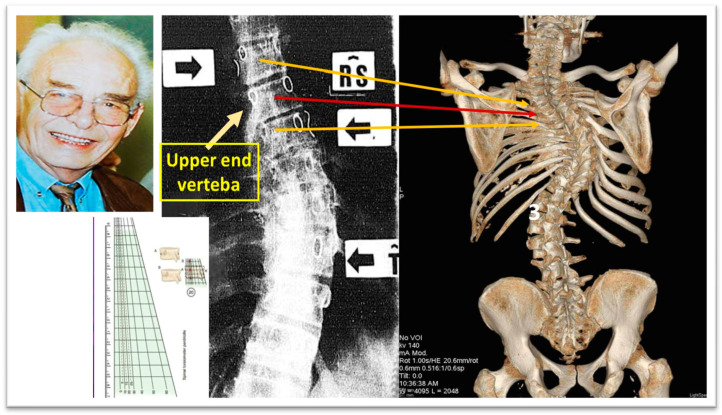
Specific Rotation, “SR”, of Perdriolle (as modified from [47]). The Perdriolle Specific Rotation is a predictor for IIS progression, namely, the sum of the two angles of rotation measured in the two vertebrae adjacent to the upper-end vertebra. IIS will progress if SR: at 2 years old > 5°, at 4 years old > 10°, at 6 years old > 20° [47].

### 4.2. Juvenile Idiopathic Scoliosis—JIS

Juvenile Idiopathic Scoliosis (JIS) emerges between the ages of 4 and 10, accounting for 10–15% of all IS cases. Curvatures measuring 30° or greater in Cobb angle are prone to worsening, with 95% of affected individuals eventually requiring surgical intervention [51]. If left untreated, these spinal deformities can lead to severe cardiopulmonary complications [52].

Early detection is crucial, particularly if scoliosis worsens within the first year of puberty. Predicting the progression of curves measuring between 21 degrees and 30 degrees during the initial two years of puberty remains challenging. However, when juvenile scoliosis surpasses 30 degrees, it tends to accelerate rapidly, with a nearly certain likelihood of requiring surgery once the curvature exceeds 40 to 45 degrees. Key indicators of progression include the curve pattern, the Cobb angle at the onset of puberty, and the rate of curve advancement [53].

### 4.3. Predictors for Progression of Adolescent Idiopathic Scoliosis—AIS

The factors typically suggested in clinical practice to predict curve progression and guide appropriate treatment for AIS are as follows: 1. age at diagnosis, 2. sex, 3. magnitude of the curve, 4. growth potential, 5. curve type, 6. curve location and Cobb angle, 7. menstrual status (for females), 8. curve pattern (single vs. double curves), and 10. family history. These represent predictors for progression of AIS patients undergoing brace treatment [54,55].

To tailor the treatment of AIS patients undergoing bracing, the Society on Scoliosis Orthopaedic and Rehabilitation Treatment (SOSORT) adapted a modified version of the Lonstein and Carlson (1984) [32], method to assess prognostic risk during the pubertal growth spurt. Additionally, the SOSORT provided recommendations on bracing based on the Risser sign for patients with IIS, JIS, and AIS, as well as in elders, as outlined in Figure 9 and Table 1 [56].

### 4.4. Predictors for Progression of Braced Adolescent Idiopathic Scoliosis (AIS)

Cheung et al. (2020) stated that when the flexibility rate exceeds 28%, the chances of preventing curve progression through bracing increase, whereas a lower flexibility rate reduces this probability [57,58].

The Supine Correction Index (SCI) is determined by dividing the correction rate by flexibility. Wong et al. (2022) found that a higher supine flexibility (18.1%), an increased correction rate (28.8%), and an SCI above 1.21 were linked to a lower risk of curve progression [59].

According to Cheung et al. (2019–2020), among AIS patients receiving bracing, those who begin treatment at a younger age and have greater vertebral wedging are more likely to experience curve progression. However, when spinal flexibility surpasses 28%, bracing is more effective in preventing progression; otherwise, the risk increases [57].

Sitoula et al. (2015) reported that 51% of patients adhered to brace treatment, and half of them still experienced curve progression. Additionally, 34% of patients who showed progression despite bracing were classified within stages SS1 and SS2 [26,54].

Khodaei et al. (2025) conducted a study to identify the predictors of brace treatment outcomes for adolescents and adults with IS. They found that eleven parameters were associated with bracing outcomes; however, most studies exhibited a moderate risk of bias. Among these parameters, only one—longer treatment duration—was predictive of better long-term SRS-22 total scores, though with limited strength of evidence. Given the generally unclear level of evidence, the study highlighted common weaknesses and emphasized the need for high-quality research to improve predictions of bracing outcomes [60]. However, a major limitation of this study is that the sole predictive parameter (longer treatment time) was derived from the SRS-22 questionnaire, which was originally developed for surgical treatment of idiopathic scoliosis rather than bracing. As a result, the use of an incorrect treatment questionnaire, despite the existence of questionnaires specifically developed for brace treatment, introduces uncertainty into the findings.

### 4.5. The Prediction of Curve Progression in Untreated Idiopathic Scoliosis During Growth

The Iowa study, which originated from research on school scoliosis screening in children, is regarded as a foundational study in understanding the progression of AIS and in developing the first predictive model for its progression. Figure 10 illustrates the incidence of progression for each curve pattern. The numbers to the right of the bars represent the number of patients with each pattern, while the numbers in parentheses indicate the percentage of cases that progressed [32]. Predictors of AIS curve progression have been found to include curve pattern and magnitude, the patient’s age at presentation, Risser sign, and menarchial status.

Curve progression during growth is very important not only because it may result in surgical treatment but it may also alter the patients’ self-perception and the sports practice of adolescents with IS [61]. Soucacos et al. (1998), through the implementation of the SSS program, examined 85,622 children. Among 1436 children re-examined for scoliosis, 14.7% had a Cobb angle greater than 10 degrees. A follow-up at 3.2 years assessed curve progression, identifying strong predictors such as sex, curve pattern, and skeletal maturity, while age and curve magnitude showed a weaker association. The incidence of curve progression, categorized by curve pattern and direction (left vs. right) for both boys and girls, is presented in Figure 11. Notably, none of the left thoracic curves exhibited progression. The incidence of progression in right thoracic curves among girls was as high as that observed for double curves, while boys with right lumbar curves had a higher progression rate than girls [62].

### 4.6. Apical Convex Rib–Vertebral Angle in AIS as a Prognostic Factor for Progression

Modi et al. 2009 recommend measuring the drooping value in the apical RVA on the convex side at regular follow-up to check for further progression of the curve and to decide on other treatment options. Measurement of the drooping value in convex RVA is equally important as that of initial convex RVA or RVAD in the literature, as shown in Figure 12 [63].

## 5. Predictors for Curve Progression at Skeletal Maturity

Examining the predictors of curve progression at skeletal maturity, research indicates that curves reaching 50 degrees are likely to continue progressing into adulthood at an average rate of 1 degree per year, as primarily shown in the Iowa studies [64].

The progression of thoracolumbar/lumbar curves is also associated with L3 and L4 tilt, particularly when exceeding 16 degrees [65]. Additionally, a Rib Index threshold of 1.915 at maturity is an indicator of rapid progression [66]. Surgical treatment should be considered for skeletally mature patients with curves exceeding 50 degrees.

## 6. Models of Prediction of Progression of IS

In recent years, there has been a growing emphasis on transdisciplinary research that integrates machine learning with clinical data to develop specialized in-house programs for predicting curve progression. However, the use of complex terminology in these studies can make them not only difficult to comprehend but also challenging to assess. Additionally, current predictive models for evaluating the progression of IS curves primarily rely on parameters obtained from 2D radiographic imaging. We believe these models require further time and validation studies to establish their reliability before they can be widely accepted and universally implemented [13,67].

One of the earliest models introduced was likely the Lonstein and Carlson prediction model for assessing the risk of AIS curve progression. This model calculates the risk factor using a specific formula, as illustrated in Figure 13 [32]. The key predictors used in this model to estimate the percentage incidence of curve progression include the Cobb angle, patient age, and the Risser sign.

Dolan et al. (2019) conducted a study on the development and validation of a prognostic model for untreated adolescent idiopathic scoliosis (AIS) using the simplified skeletal maturity system. Their model incorporated predictors such as Cobb angle, age, sex, curve type, triradiate cartilage status, and skeletal maturity stage. As their final functional model, they recommended using skeletal maturity stage, the Cobb angle, and curve type [67].

A linear mixed-effects model was applied to 2317 patients with idiopathic scoliosis (IS) aged 6 to 25 years. The predictors included age, sex, maximum Cobb angle, Risser stage, and curve type. These validated models demonstrated good accuracy in predicting future Cobb angles in untreated IS across the entire growth spectrum [13].

Wan et al. (2024) reported a 3D prediction model based on 2D radiological reconstruction. The risk of bias was assessed using the Quality in Prognostic Studies (QUIPS) tool and the Appraisal Tool for Cross-Sectional Studies (AXIS). Additionally, the level of evidence for each predictor was rated using the Grading of Recommendations, Assessment, Development, and Evaluations (GRADE) approach. Key predictors included the torsion index (TI), apical vertebral rotation (AVR), thoracic hypokyphosis, sagittal wedging, and the initial Cobb angle. In mild curves, TI and AVR were identified as predictors of curve progression, with TI > 3.7° and AVR > 5.8° considered significant thresholds [68].

In recent years, 3D reconstruction of biplanar radiographs has gained increasing interest due to its validated accuracy and reproducibility [10,69]. Despite providing extensive quantitative data, commercially available biplanar reconstruction programs still require substantial manual effort to map spinal landmarks before automated measurements can be performed [70].

Dufvenberg et al. (2024), conducted a prospective cohort study involving 127 patients, aiming to develop a prognostic model for assessing the risk of curve progression in AIS. The study utilized the Cox proportional hazards (CoxPH) regression survival model for model development and validation, comparing its performance with machine learning models using a 66.6/33.3 train/test data split, as well as the patient-reported Spinal Appearance Questionnaire (pSAQ). The models were adjusted for treatment exposure, and 34 candidate prognostic variables were evaluated. The final prognostic model, incorporating the Risser stage, Cobb angle, pSAQ, and menarche, demonstrated acceptable discriminative ability in predicting curve progression of more than 6 degrees in Cobb angle. The inclusion of patient-reported pSAQ may have clinical significance in forecasting curve progression. The CoxPH model (Cox, 1972) remains the most widely used multivariate approach for analyzing survival time data in medical research, describing the relationship between event incidence (expressed via the hazard function) and a set of covariates [71].

Nault et al. (2020), developed a predictive model for AIS progression based on 3D spinal parameters assessed at the initial visit. A general linear model with backward selection was employed, using the final Cobb angle—measured either just before surgery or at skeletal maturity—as the outcome variable. The analysis, conducted on 172 patients, identified significant predictors, including initial skeletal maturation, curve type, the frontal Cobb angle, the angle of the plane of maximal curvature, and 3D disc wedging (T3–T4, T8–T9) [11].

Alfraihat et al. (2022) utilized a random forest machine learning (ML) model to predict AIS curve progression. The study incorporated radiographic features previously associated with curve progression, applying Sequential Backward Floating Selection (SBFS) to determine the most predictive subset of features. The key predictors identified were the Cobb angle, spinal flexibility, initial lumbar lordosis angle, initial thoracic kyphosis angle, age at last visit, number of levels involved, and Risser “+” stage at the first visit. However, the relative importance and optimal combination of these predictive factors remain uncertain [72].

## 7. Rib Index: A Thoracic Deformity Parameter as Predictor of Progression

The Rib Index (RI) method was first introduced in 1999 [73] and was later associated with scoliogenesis [74,75]. It was developed from the Double Rib Contour Sign (DRCS) to evaluate rib hump deformity (RHD) in individuals with IS, providing a reliable and consistent approach for assessing RHD using lateral radiographs, as shown in Figure 14 [76,77].

Clinically, beyond documenting the deformity [78], the RI method has been utilized for evaluating physiotherapeutic-specific scoliosis exercises (PSSE) [79], assessing brace treatment [80,81], and conducting pre- and postoperative evaluations of thoracic deformity correction across various instrumentation techniques [77,82,83,84,85,86,87,88].

Upon reviewing the literature, only one comprehensive review was found that analyzed rib cage deformity parameters in scoliosis and provided a detailed description of them [89].

However, there is limited information regarding the relationship between these parameters and their impact on spinal deformity. A review of the existing studies identified only one publication addressing thoracic deformity parameters used to predict progression in skeletally mature AIS curves measuring 40–50 degrees. Shea et al. (2024) [66], in their study “Rib Index Prognoses Accelerated Deterioration in Skeletally Mature AIS Curves of 40–50° Using Uniplanar Radiographic Measures”, reported that a Rib Index threshold of 1.915 at maturity was associated with rapid progression.

Further research on segmental rib index (SRI) in IS rib cages [90] revealed that at any level from T1 to T12, a segmental RI value of 1.45–1.50 or higher primarily indicates a significantly asymmetrical rib cage deformity, suggesting notable asymmetric rib growth at that spinal level. This RI threshold represents an increasing and progressive rib cage deformity [86]. The term “pattern of segmental RI asymmetry” refers to the number of rib levels (from T1 to T12) exhibiting this severe asymmetry (≥1.45–1.50) and may serve as a predictor of T and TL curve progression.

Another intriguing implication is that spinal deformity results from rib cage asymmetry, aligning with the late Prof. John Sevastikoglou’s (Sevastik’s) thoracospinal concept [91].

The perspective that rib deformation is not closely linked to vertebral rotation and that it precedes spinal deformation, particularly in thoracic curves, is primarily supported by the research of Professor J. Sevastik [92,93,94,95,96,97,98,99,100,101,102,103,104] and other published studies [105,106].

Theoretically, if rib cage deformation were solely due to vertebral rotation, then surgically aligning and derotating the vertebral column would fully correct the rib cage deformity postoperatively. However, clinical practice shows this is not the case. Therefore, analyzing published postoperative outcomes of IS scoliosis correction on rib cage deformities can provide an objective answer to this question [66,86,87]. This suggests that rib hump deformity (RHD) is more likely a result of asymmetric rib development rather than vertebral rotation, contrary to common belief [105].

We recommend using the rib index (RI) and segmental rib index (SRI) as reliable predictors of IS curve progression. This recommendation is based on findings from both preoperative AIS cases and postoperative cases with progression [66,86,87].

Igoumenou et al. (2021) [87], reported preoperative RI values in different surgical constructs: full pedicle screw constructs (group A), hybrid constructs (group B), and Harrington rods (group C). The preoperative mean RI was 2.05 ± 0.23 in group A, 1.93 ± 0.27 in group B, and 2.06 ± 0.25 in group C. Postoperatively, the RI values were 1.37 ± 0.12 (group A), 1.38 ± 0.12 (group B), and 1.61 ± 0.26 (group C). The mean RI correction was 32.7 ± 5.1% for group A, 28.7 ± 5.5% for group B, and 26.8 ± 6.9% for group C [82,83]. This statement implies that the postoperative average correction of the RI in this study clearly illustrates that spinal alignment and vertebral derotation achieved through instrumental posterior spinal fusion are not sufficient to fully correct the rib hump or overall trunk rotation, regardless of the surgical technique used.

Consequently, it is suggested that the primary factor contributing to rotational deformity in AIS is the asymmetrical growth of rib pairs. These findings also suggest that the rib cage deformity observed in preoperative IS patients, as measured by the RI and SRI, could serve as a predictor of spinal curve progression. In other words, when an IS patient reaches the stage where surgical intervention is indicated, the RI reflects the degree of rib cage deformity present. Therefore, this level of rib cage deformity could reasonably be considered a predictor for the necessity of surgical treatment.

**Figure 14 medsci-13-00062-f014:**
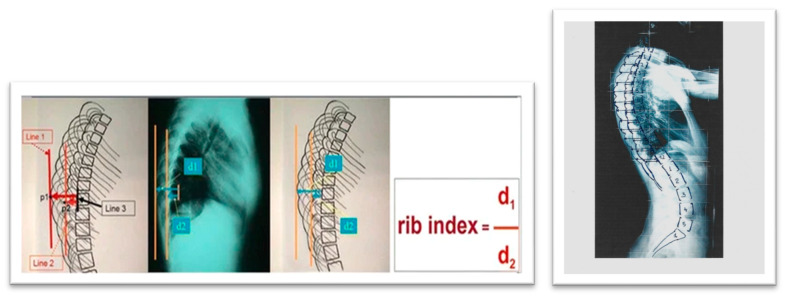
Rib Index and Segmental Rib Index [76,90].

While comparing the RI and SRI with other methods for predicting scoliosis progression would be insightful, it is not currently scientifically feasible due to the lack of relevant studies. Nonetheless, we acknowledge this absence as a limitation of our study.

## 8. Critical Opinion

The rationale for proposing the RI and SRI as useful predictors is based on the following consideration: The existing literature on IS includes publications reporting surgical outcomes in patients with widely accepted indications for surgical intervention—that is, patients with advanced IS [66,77,82,85,86,87,91]. In these studies, the preoperative RI was compared to the postoperative RI. If IS has reached the stage where surgery is indicated, then the RI, which reflects the thoracic transverse plane deformity, must be reliably abnormal. This suggests that the RI can serve as a valid indicator of deformity progression, as other established indicators.

## 9. Conclusions

Prediction of curve progression was always a challenge for physicians who treated children with idiopathic scoliosis. Cobb angle and skeletal maturity were traditionally used in decision making, while the age of scoliosis diagnosis was a prognostic factor for progression of the curve and determined which treatment was suitable for each individual. The present paper summarizes the current knowledge of factors that are involved in curve progression in idiopathic scoliosis and cites the numerous models that have been developed for prediction of IS progression. It also highlights the importance of the thorax in scoliotic deformity and introduces, for the first time, two factors that describe the deformity of the rib cage, namely, the Rib Index and the Segmental Rib Index, as predictors of IS progression.

## Figures and Tables

**Figure 1 medsci-13-00062-f001:**
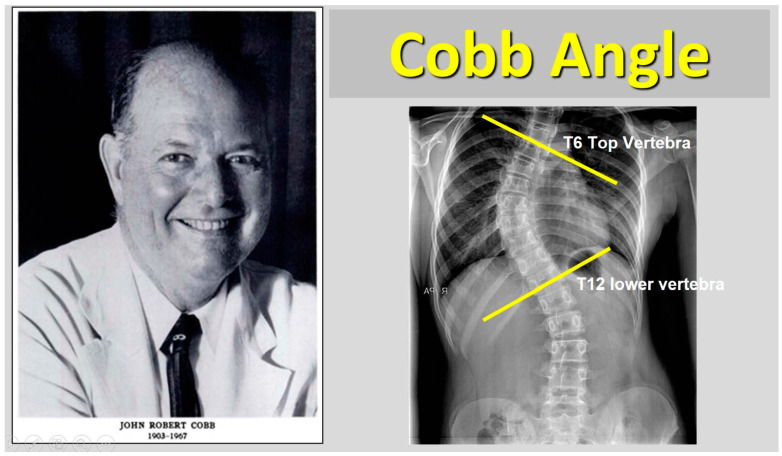
The Cobb angle is the sum of upper- and lower-end vertebra tilt angles.

**Figure 2 medsci-13-00062-f002:**
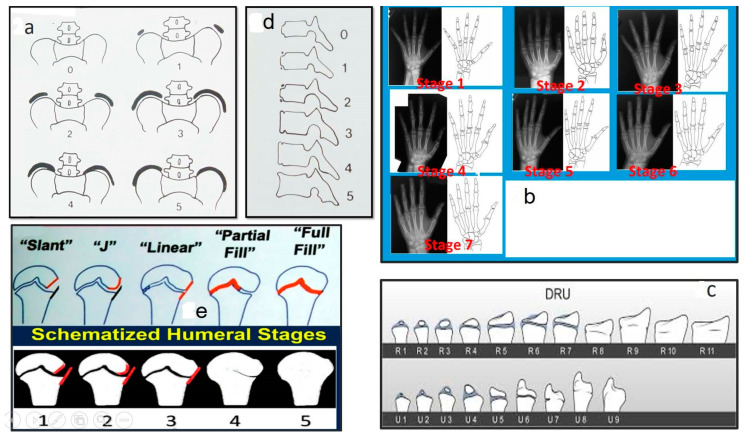
The various employed methods/techniques to evaluate the level of skeletal maturity, modified from original published figures (as modified from [25,26,27,28,29,30]). (**a**) Risser’s method, known as Risser sign, (**b**) Sanders maturity score system, (**c**) distal radius and ulna (DRU) classification, (**d**) ossification of the vertebral epiphyseal rings, (**e**) ossification of the humeral head.

**Figure 3 medsci-13-00062-f003:**
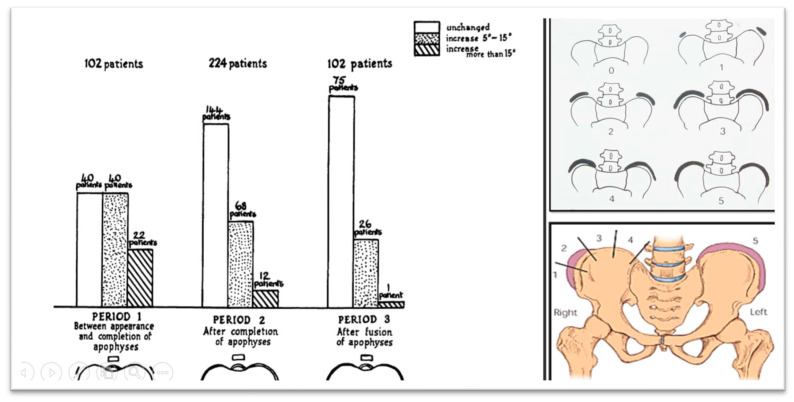
The Iliac apophysis and the evolution of curves in scoliosis (as modified from [31]).

**Figure 4 medsci-13-00062-f004:**
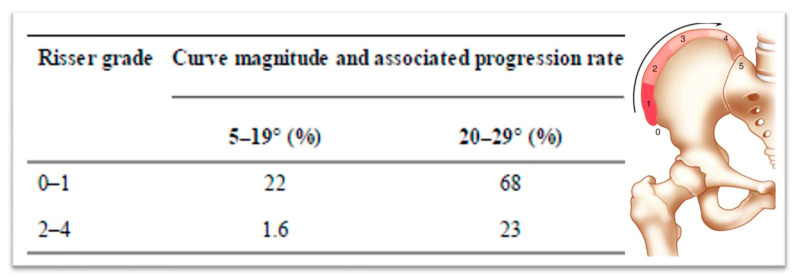
The likelihood of curve progression by the Risser sign grade and curve magnitude (as modified from [32]).

**Figure 5 medsci-13-00062-f005:**
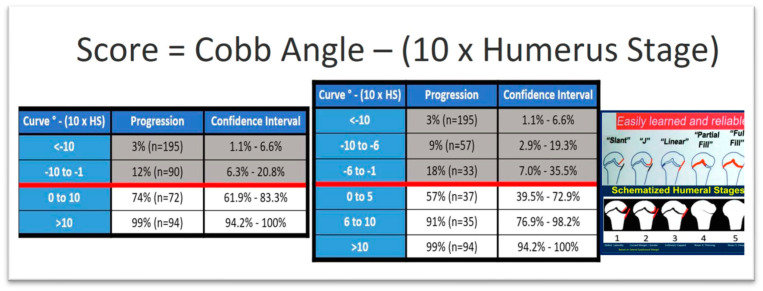
Risk of progression to surgical range based on the above-depicted formula which determines the score (modified from Li et al. [33]).

**Figure 9 medsci-13-00062-f009:**
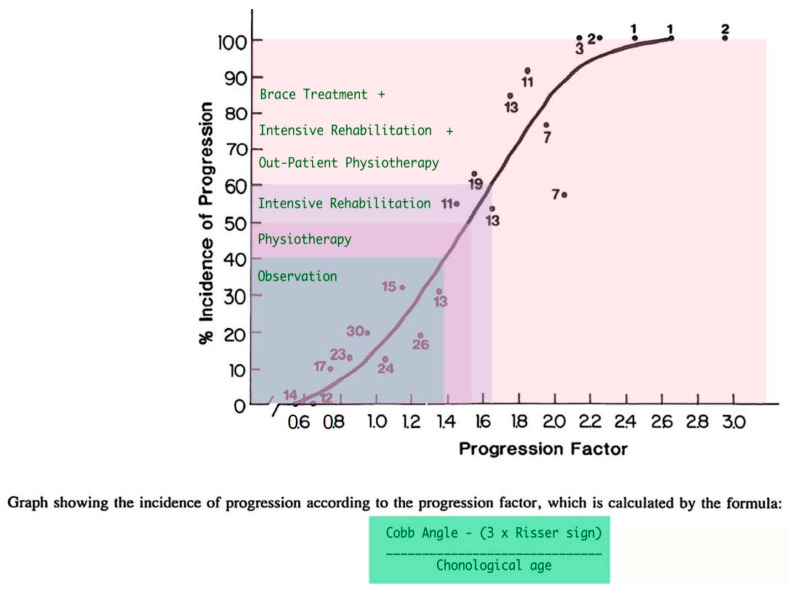
The estimation of the prognostic risk during the pubertal growth spurt for tailoring brace treatment (as modified from [56]).

**Figure 10 medsci-13-00062-f010:**
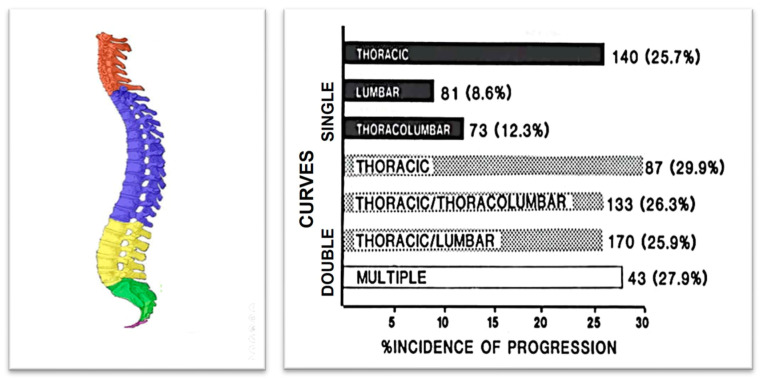
The incidence of progression for each curve pattern (as modified from [32]).

**Figure 11 medsci-13-00062-f011:**
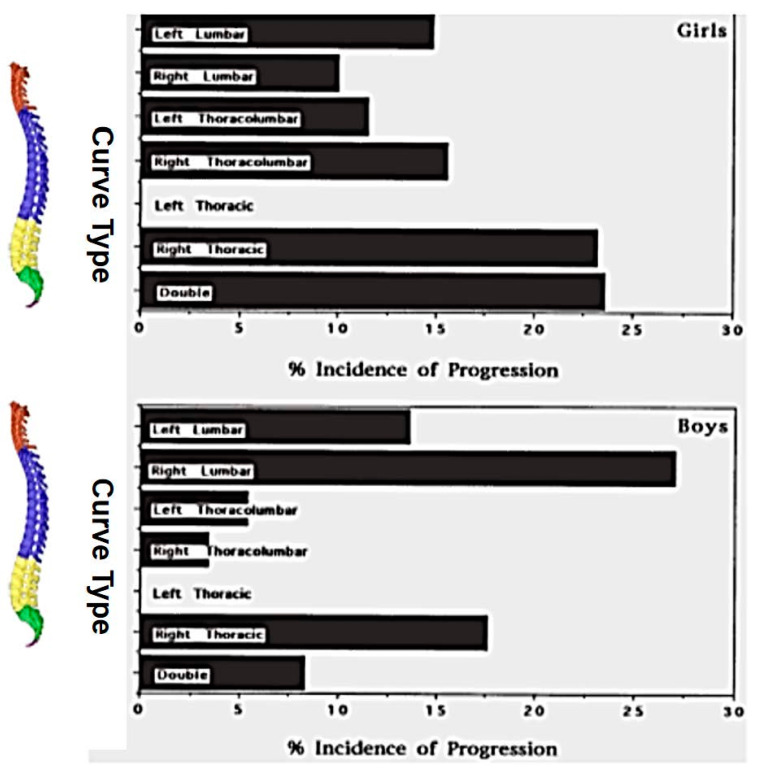
The incidence of curve progression based on curve pattern and direction (left vs. right) for boys and girls (as modified from [62]).

**Figure 12 medsci-13-00062-f012:**
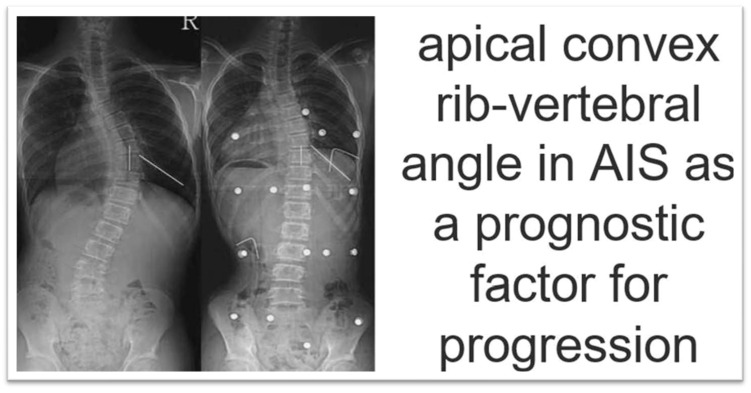
Measurement of the drooping value in convex RVA is equally important as that of initial convex RVA or RVAD in the literature (as modified from [63]).

**Figure 13 medsci-13-00062-f013:**
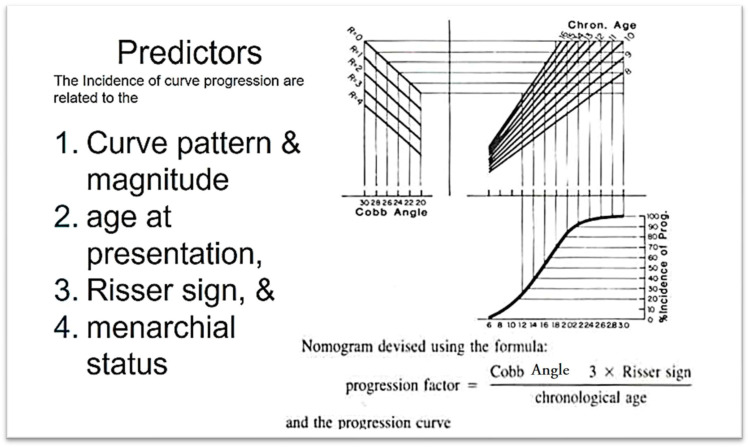
The nomogram of assessing the progression factor of an untreated IS curve (as modified from [32]).

**Table 1 medsci-13-00062-t001:** SOSORT recommendation on bracing by Risser sign in IIS, JIS, AIS, adulthood and elderly populations, the numbers indicate months (as modified from [56]).

		Cobb Degrees	0–10 + hump	11–15	16–20	21–25	26–30	31–35	36–40	41–45	46–50	Over 50
**Infantile**		Min	Ob6	Ob6	Ob3	SSB	SSB	SSB	SSB	SSB	PTRB	FTRB
		Max	Ob3	Ob3	PTRB	FTRB	FTRB	FTRB	FTRB	FTRB	Su	Su
Juvenile		Min	Ob3	Ob3	Ob3	SSB	SSB	SSB	PTRB	PTRB	PTRB	FTRB
		Max	PSE	PSE	PTRB	FTRB	FTRB	FTRB	FTRB	FTRB	Su	Su
Adolescent	Risser 0	Min	Ob6	Ob6	Ob3	PSE	PSE	SSB	PTRB	PTRB	PTRB	FTRB
		Max	Ob3	PSE	PTRB	FTRB	FTRB	FTRB	FTRB	FTRB	Su	Su
	Risser 1	Min	Ob6	Ob6	Ob3	PSE	PSE	SSB	PTRB	PTRB	PTRB	FTRB
		Max	Ob3	PSE	PTRB	FTRB	FTRB	FTRB	FTRB	FTRB	Su	Su
	Risser 2	Min	Ob8	Ob6	Ob3	PSE	PSE	SSB	SSB	SSB	SSB	FTRB
		Max	Ob6	PSE	PTRB	FTRB	FTRB	FTRB	FTRB	FTRB	Su	Su
	Risser 3	Min	Ob12	Ob6	Ob6	Ob6	PSE	SSB	SSB	SSB	SSB	FTRB
		Max	Ob6	PSE	PTRB	FTRB	FTRB	FTRB	FTRB	FTRB	Su	Su
	Risser 4	Min	No	Ob6	Ob6	Ob6	Ob6	Ob6	Ob6	Ob6	SSB	FTRB
		Max	Ob12	PSE	PTRB	FTRB	FTRB	FTRB	FTRB	FTRB	Su	Su
	Risser 4–5	Min	No	Ob6	Ob6	Ob6	Ob6	Ob6	Ob6	Ob6	SSB	FTRB
		Max	Ob12	PSE	PTRB	FTRB	FTRB	FTRB	FTRB	FTRB	Su	Su
Adult	No pain	Min	No	No	No	No	No	No	No	No	Ob12	Ob12
		Max	Ob12	Ob12	Ob12	Ob12	Ob12	Ob12	Ob12	Ob12	Ob6	Ob6
	Chronic Pain	Min	No	PSE	PSE	PSE	PSE	PSE	PSE	PSE	PSE	PSE
		Max	PTRB	PTRB	PTRB	PTRB	PTRB	Su	Su	Su	Su	Su
Elderly	No pain	Min	No	No	No	No	No	No	No	No	Ob12	Ob12
		Max	Ob12	Ob12	Ob12	Ob12	Ob12	Ob12	Ob12	Ob12	Ob6	Ob6
	Chronic Pain	Min	No	PSE	PSE	PSE	PSE	PSE	PSE	PSE	PSE	PSE
		Max	PTRB	PTRB	PTRB	PTRB	PTRB	PTRB	PTRB	PTRB	Su	Su

## Data Availability

No new data were created or analyzed in this study.

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
