# Peer review of "Idiopathic Scoliosis Progression: Presenting Rib and Segmental Rib Index as Predictors—A Literature Review"

_medsci, 2025, doi:10.3390/medsci13020062_

Round 1
Reviewer 1 Report
Comments and Suggestions for Authors
The authors provide a very comprehensive overview of risk factors for progression of scoliosis. They cover all major categories of scoliosis including infantile to idiopathic scoliosis.
The review may be enhanced by the following suggestions.
In figure 2, it is quite difficult to discern the various employed techniques to assess skeletal maturity. Technically, please separate the figure into parts (like 2a, 2b etc) to show the various methods. As a literature review, it is important for the figures to be clear but there are no clear indicators for which method is which in the figure. For example, placing the Risser’s method text under the part of the figure properly and specificialy labelled with legends showing the Risser sign would be a better way to arrange the figure.
Figure 5, it is arranged akin to a powerpoint slide. Please remove the text in the lower part and put them into a legend.
Rib index appears as a very useful predictor. Could the authors provide more details like: How does Rib Index compare with other methods as a predictor of scoliosis progression, as you have recommended using Rib index as the reliable predictor of scoliosis progression.
Author Response
The authors provide a very comprehensive overview of risk factors for progression of scoliosis. They cover all major categories of scoliosis including infantile to idiopathic scoliosis. The review may be enhanced by the following suggestions.
Dear reviewer thanks you for your time and recommendations
Our response to your comments:
In figure 2, it is quite difficult to discern the various employed techniques to assess skeletal maturity. Technically, please separate the figure into parts (like 2a, 2b etc) to show the various methods.
Dear Reviewer thank you.
We have added labels to each part of the figure (e.g., 2a, 2b, etc.) as suggested, but we did not separate the individual parts. The revised layout in Figure 2 does not require formal permission from the original authors or publishers, as long as proper attribution is provided—which we have included in the figure labels. This approach addresses the reviewer’s comment while simplifying the process. However, if this is not acceptable, we are willing to remove the figure, although we believe it is valuable for readers."
As a literature review, it is important for the figures to be clear but there are no clear indicators for which method is which in the figure.
For example, placing the Risser’s method text under the part of the figure properly and specifically labelled with legends showing the Risser sign would be a better way to arrange the figure.
Dear Reviewer, thank you. We chose not to include detailed descriptions of each method, as our primary objective is to focus on our main contribution: introducing the Rib Index and Segmental Rib Index as predictors of progression in idiopathic scoliosis. The other methods mentioned are described concisely and can be easily accessed through the references cited in the literature section of our submission.
Figure 5, it is arranged akin to a PowerPoint slide. Please remove the text in the lower part and put them into a legend.
Dear Reviewer thank you. We removed the text in the lower part and put them into the legend.
Rib index appears as a very useful predictor. Could the authors provide more details like: How does Rib Index compare with other methods as a predictor of scoliosis progression, as you have recommended using Rib index as the reliable predictor of scoliosis progression.
Dear Reviewer thank you.
The rationale for proposing the RI and SRI as useful predictors is based on the following consideration: Existing literature on idiopathic scoliosis (IS) includes publications reporting surgical outcomes in patients with widely accepted indications for surgical intervention—that is, patients with advanced IS. In these studies, the preoperative RI was compared to the postoperative RI. If IS has reached the stage where surgery is indicated, then the RI, which reflects the thoracic transverse plane deformity, must be reliably abnormal. This suggests that the RI can serve as a valid indicator of deformity progression, comparable to other established indicators. While the idea of comparing the RI and SRI with other methods for predicting scoliosis progression is insightful, such a comparison is not currently scientifically feasible due to the absence of relevant studies. However, we appreciate the reviewers’ thoughtful comment, which presents an excellent idea for future research. We will also acknowledge this as a limitation of our study in the discussion section.
Reviewer 2 Report
Comments and Suggestions for Authors
- The novelty and importance of this review should be carefully discussed in Introduction and Conclusion.
- The previous review papers in this topic should be discussed in Introduction.
- Part 1 is introduction but not “Cobb Angle – Rib Hump”. The introduction for Cobb Angle – Rib Hump should be revised with more discussions and the copyright permission for Figure 1 should be provided.
- The manuscript should be submitted by using the template of this journal.
- No conclusion for the content of this review was found.
- The paragraphs should be integrated and the logicality should be improved for good readability. Many paragraphs only include one or two sentence.
Author Response
Dear Reviewer than you for your time and recommendations.
The novelty and importance of this review should be carefully discussed in Introduction and Conclusion.
- The previous review papers in this topic should be discussed in Introduction. Dear Reviewer, thank you. Three additional papers were included in the Introduction section as suggested. Papers by Peterson et al and Lonstein et al are considered milestones in scoliosis literature.
- Part 1 is introduction but not “Cobb Angle – Rib Hump”. The introduction for Cobb Angle – Rib Hump should be revised with more discussions and the copyright permission for Figure 1 should be provided.
Dear Reviewer, thank you. We chose not to include detailed descriptions of each method, as our primary objective is to focus on our main contribution: introducing the Rib Index and Segmental Rib Index as predictors of progression in idiopathic scoliosis. The other methods mentioned are described concisely and can be easily accessed through the references cited in the literature section of our submission. Figure 1 was modified using an anonymized personal case radiograph, and I hope this is acceptable to the reviewer. The image of Cobb’s face was sourced from the internet, and there was no indication that permission was required for its use. However, if this is deemed inappropriate, we are willing to remove that part of the figure."
- The manuscript should be submitted by using the template of this journal. Dear Reviewer, thank you. We consider tha the lay-out was shorted out bu the editorial office personel.
- No conclusion for the content of this review was found.
Dear Reviewer, thank you. Conclusions were added
- The paragraphs should be integrated and the logicality should be improved for good readability. Many paragraphs only include one or two sentence.
Dear Reviewer, thank you. Paragraphs were integrated and their layout improved as suggested.
Reviewer 3 Report
Comments and Suggestions for Authors
The scientific article "Idiopathic Scoliosis Progression Predictors: Presenting Rib and Segmental Rib Index as Predictors – A Literature Review" aimed to provide a concise selective representative overview of the predictor factors for progression in Idiopathic Scoliosis. I can make the following comments:
1) Change title to "Idiopathic Scoliosis Progression: Presenting Rib and Segmental Rib Index as Predictors – A Literature Review"
2) Pay attention to the formatting of the manuscript in the template. The affiliation numbers should be after the authors' names. There are more inaccuracies in the text. Please review it.
3) The abstract must contain the conclusions of the manuscript.
4) Abbreviations, when first cited in the text, must contain their full meaning. Please review the text. Ex: line 97 RVA
5) The figures used do not appear to be original. Do the authors have your authorization for publication?
6) I missed a brief anatomical description of the bone structures described, such as the spine.
7) At the end of the manuscript, before the conclusions, add a "critical opinion" section with a greater demonstration of the authors' opinions and clinical contributions to the area of ​​knowledge.
8) Add the limitations of the study.
9) In context, the manuscript presents a relevant topic in the area. After the necessary corrections, it may be eligible for publication.
Author Response
Dear Reviewer than you for your time and recommendations.
The scientific article "Idiopathic Scoliosis Progression Predictors: Presenting Rib and Segmental Rib Index as Predictors – A Literature Review" aimed to provide a concise selective representative overview of the predictor factors for progression in Idiopathic Scoliosis. I can make the following comments:
- Change title to "Idiopathic Scoliosis Progression: Presenting Rib and Segmental Rib Index as Predictors – A Literature Review"
Dear Reviewer, thank you. The titled was chanced as recommended.
2) Pay attention to the formatting of the manuscript in the template. The affiliation numbers should be after the authors' names. There are more inaccuracies in the text. Please review it. Dear Reviewer, thank you. Formatting of the text was revised and the affiliation numbers were placed after the author’s names as suggested.
3) The abstract must contain the conclusions of the manuscript.
Dear Reviewer, thank you. The conclusions were added to the abstract.
4) Abbreviations, when first cited in the text, must contain their full meaning. Please review the text. Ex: line 97 RVA. Dear Reviewer, thank you. All abbreviations were carefully explained when first cited in the text as suggested.
5) The figures used do not appear to be original. Do the authors have your authorization for publication? Dear Reviewer, thank you. On this issue please refer to our response to reviewers 1 and 2. Additionally many other figures in the submissions are from our cited publications for which we obtain the copyright as these publications are in open access journals. We note that these are modifications from our citations.
6) I missed a brief anatomical description of the bone structures described, such as the spine. Dear Reviewer, thank you A brief anatomical description of the spine and a definition of a scoliotic curve was added in Section 2 is added as suggested.
7) At the end of the manuscript, before the conclusions, add a "critical opinion" section with a greater demonstration of the authors' opinions and clinical contributions to the area of ​​knowledge. Dear Reviewer, thank you. A "critical opinion" section was added.
8) Add the limitations of the study. Dear Reviewer, thank you. The limitation of the study was added.
9) In context, the manuscript presents a relevant topic in the area. After the necessary corrections, it may be eligible for publication. Dear Reviewer, thank you for your opinion.
Reviewer 4 Report
Comments and Suggestions for Authors
Introduction:
The introduction provides a solid context regarding the predictors of progression in idiopathic scoliosis, but it lacks initial quantitative data on incidence or prevalence. There is no introduction to the article’s quantitative approach or to the strategy for selecting or synthesizing statistical evidence. It would be beneficial to include estimated epidemiological values for progression and prevalence by age group or severity. Furthermore, a rationale for the use of rib indexes (RI and SRI) as potential quantitative predictors should be presented, preferably referencing studies that evaluate their accuracy.
Methodology:
This section mentions various predictors (e.g., Risser, Cobb, DRU, Sanders), yet it does not provide a direct statistical comparison among these methods, nor does it include measures such as sensitivity or specificity. A comparative table between the methods (e.g., Risser vs. Sanders vs. DRU) incorporating values such as AUC, sensitivity, and specificity—if available—would enhance the methodological clarity. Additionally, it would be important to clarify which studies conducted internal or external validations of the predictive models.
Results:
The most impactful statistical findings could be summarized, for instance, by reporting hazard ratio (HR) values or predictive accuracy (above 80%?) of the models. Future studies might consider comparing traditional models with machine learning approaches, using metrics such as AUC or ROC curves. The use of radiographic indices as predictors requires formal statistical validation—logistic regression, ROC curve analysis, likelihood ratios, and positive/negative predictive values.
The absence of these data renders the proposal interesting, yet still speculative from a statistical standpoint.
Conclusion:
The conclusion does not address statistical limitations, such as the lack of empirical data supporting the indices as independent predictors. It would be appropriate to include explicit caveats indicating that RI/SRI still lack formal predictive validation and should be regarded as complementary markers in future studies.
Author Response
Introduction:
The introduction provides a solid context regarding the predictors of progression in idiopathic scoliosis, but it lacks initial quantitative data on incidence or prevalence. There is no introduction to the article’s quantitative approach or to the strategy for selecting or synthesizing statistical evidence. It would be beneficial to include estimated epidemiological values for progression and prevalence by age group or severity. Furthermore, a rationale for the use of rib indexes (RI and SRI) as potential quantitative predictors should be presented, preferably referencing studies that evaluate their accuracy.
Dear Reviewer,
Thank you for your comment. As mentioned in our responses to the other reviewers, the title of the article was revised in accordance with the suggestion made by one of them. Consequently, the data you refer to in your comment can be found in the cited references, as the article provides a concise summary of the methods used to assess scoliosis progression to date. Our primary focus is on the RI and SRI as predictors of progression. Additionally, we have included a "Critical Opinion" section in the manuscript that directly addresses your concerns and discusses the limitations of the study.
Methodology:
This section mentions various predictors (e.g., Risser, Cobb, DRU, Sanders), yet it does not provide a direct statistical comparison among these methods, nor does it include measures such as sensitivity or specificity. A comparative table between the methods (e.g., Risser vs. Sanders vs. DRU) incorporating values such as AUC, sensitivity, and specificity—if available—would enhance the methodological clarity. Additionally, it would be important to clarify which studies conducted internal or external validations of the predictive models.
Dear Reviewer, Thank you for your comment. Please see our first response.
Results:
The most impactful statistical findings could be summarized, for instance, by reporting hazard ratio (HR) values or predictive accuracy (above 80%?) of the models. Future studies might consider comparing traditional models with machine learning approaches, using metrics such as AUC or ROC curves. The use of radiographic indices as predictors requires formal statistical validation—logistic regression, ROC curve analysis, likelihood ratios, and positive/negative predictive values.
The absence of these data renders the proposal interesting, yet still speculative from a statistical standpoint.
Dear Reviewer, Thank you for your comment. Please see our first response.
Conclusion:
The conclusion does not address statistical limitations, such as the lack of empirical data supporting the indices as independent predictors. It would be appropriate to include explicit caveats indicating that RI/SRI still lack formal predictive validation and should be regarded as complementary markers in future studies.
Dear Reviewer than you for your comment. In the "Critical Opinion" section of our manuscript that indirectly addresses your concerns and discusses the limitations of the study.
Reviewer 5 Report
Comments and Suggestions for Authors
The argument of this paper is interesting. It describes Idiopathic Scoliosis progression predictors according to the scientific literature.
Nevertheless, there are some methodological concerns to resolve, so the article should be better assessed by the authors.
The layout is confusing. For example, the title line space should be corrected. Similarly, there are different styles along the main text. Please check that all the text has a layout in line with this Journal editorial rules.
Be careful also about the use of figures when they are not your own.
Introduction is fine and well addressed.
The remaining text should be better structured and re-organized. A materials and methods section is mandatory. It is necessary a separate section for this. You should here explain which type of review you carried out. Is this a systematic review? If yes, as you know, it is necessary a Prospero registration, the quality assessment and the risk of bias assessment. Is it a scoping review? Is it an overview? Also in this case, you should describe the work organization among the involved investigators, you must define the search string you used and the scientific databases you used. A Prisma flowchart for the search work should be provided. Also a summary table including the studies you selected for each research item could be useful for a better readibility of the article. Similarly, a prisma checklist could be useful for assessing the whole structure of the paper.
As the findings, they are interesting, but you should also enlarge the discussion describing the importance of self-perception and sports practice for adolescent suffering from idiopatic scoliosis. To do that, I suggest the following reference:
Notarnicola A, Farì G, Maccagnano G, Riondino A, Covelli I, Bianchi FP, Tafuri S, Piazzolla A, Moretti B. Teenagers’ perceptions of their scoliotic curves. an observational study of comparison between sports people and non- sports people. Muscles Ligaments Tendons J [Internet]. 2019;9(2):225-35.
A separate paragraph for conclusions is needed.
Best regards and good luck
Author Response
Dear Reviewer, thank you for your time and recommendations.
The argument of this paper is interesting. It describes Idiopathic Scoliosis progression predictors according to the scientific literature.
Nevertheless, there are some methodological concerns to resolve, so the article should be better assessed by the authors.
The layout is confusing. For example, the title line space should be corrected. Similarly, there are different styles along the main text. Please check that all the text has a layout in line with this Journal editorial rules.
Dear Reviewer than you, the layout was modified and corrected.
Be careful also about the use of figures when they are not your own.
Dear Reviewer than you This comment was addressed in our response to other 4 reviewers.
Introduction is fine and well addressed. Dear Reviewer than you.
The remaining text should be better structured and re-organized. A materials and methods section is mandatory. It is necessary a separate section for this. You should here explain which type of review you carried out. Is this a systematic review? If yes, as you know, it is necessary a Prospero registration, the quality assessment and the risk of bias assessment. Is it a scoping review? Is it an overview? Also in this case, you should describe the work organization among the involved investigators, you must define the search string you used and the scientific databases you used. A Prisma flowchart for the search work should be provided. Also a summary table including the studies you selected for each research item could be useful for a better readibility of the article. Similarly, a prisma checklist could be useful for assessing the whole structure of the paper.
Dear Reviewer than you for your comments. Actually this is a concise report of the used methods of assessing the IS curve progression and we propose our suggestion as it is explained in the critical opinion section of the submission. Actually it is a narrative review which reports what is known in the literature about progression factors of idiopathic scoliosis and introduces the role of rib index and segmental rib index as predictors of such progression. It is not a systematic review, and this is the reason why the authors did not follow the rules of writing such an article (registration with Prospero, Prisma flowchart, summary table of all the articles used). Additionally, as a narrative review a Materials and Methods section is not applicable.
As the findings, they are interesting, but you should also enlarge the discussion describing the importance of self-perception and sports practice for adolescent suffering from idiopathic scoliosis. To do that, I suggest the following reference:
Notarnicola A, Farì G, Maccagnano G, Riondino A, Covelli I, Bianchi FP, Tafuri S, Piazzolla A, Moretti B. Teenagers’ perceptions of their scoliotic curves. an observational study of comparison between sports people and non- sports people. Muscles Ligaments Tendons J [Internet]. 2019;9(2):225-35.
The reference by Notarnicola et al was added as suggested by the reviewer.
A separate paragraph for conclusions is needed. Dear Reviewer, thank you. Conclusions as a separate paragraph were added
Best regards and good luck Dear Reviewer, thank you very much.
Round 2
Reviewer 2 Report
Comments and Suggestions for Authors
The conclusion is too simple and the sentences are the same as those in Abstract.
Author Response
Dear reviewer,
we added to the conclusions all the information that the current paper discusses as suggested. The current form uses different sentences from the Abstract.
Reviewer 3 Report
Comments and Suggestions for Authors
No comments. Thanks
Author Response
Dear reviewer, thank you for your valuable comments for our paper.
Reviewer 5 Report
Comments and Suggestions for Authors
The paper is now suitable for publication, so no further corrections are needed
Author Response
Dear reviewer, thank you for your valuable comments and the acceptance of our paper.